# Optimization Conditions of Malachite Green Adsorption onto Almond Shell Carbon Waste Using Process Design

**DOI:** 10.3390/molecules29010054

**Published:** 2023-12-21

**Authors:** Faiza Chouli, Abdelrahman Osama Ezzat, Lilia Sabantina, Abdelghani Benyoucef, Abdelhafid Zehhaf

**Affiliations:** 1LMAE Laboratory, Department of Process Engineering, Faculty of Science and Technologies, Mascara University, Mascara 29000, Algeria; chouli.faiza@univ-mascara.dz; 2Department of Chemistry, College of Sciences, King Saud University, Riyadh 11451, Saudi Arabia; aezzat@ksu.edu.sa; 3Department of Apparel Engineering and Textile Processing, Berlin University of Applied Sciences—HTW, 12459 Berlin, Germany; 4Department of Textile and Paper Engineering, Polytechnic University of Valencia, E-03801 Alcoy, Spain; 5LSTE Laboratory, Department of Process Engineering, Faculty of Science and Technologies, Mascara University, Mascara 29000, Algeria; 6Laboratory of Process Engineering and Chemistry Solution, Department of Process Engineering, Faculty of Science and Technologies, Mascara University, Mascara 29000, Algeria; a.zehhaf@univ-mascara.dz

**Keywords:** almond shell, malachite green, adsorption, thermodynamic study, kinetic study, design process

## Abstract

Almond shell-based biocarbon is a cheap adsorbent for the removal of malachite green, which has been investigated in this work. FT-IR, DRX, and BET were used to characterize almond shell-based biocarbon. The nitrogen adsorption-desorption isotherms analysis results showed a surface area of 120.21 m^2^/g and a type H4 adsorption isotherm. The parameters of initial dye concentration (5–600 mg.L^−1^), adsorbent mass (0.1–0.6 mg), and temperature (298–373 K) of adsorption were investigated. The experiments showed that the almond shell could be used in a wide concentration and temperature range. The adsorption study was fitted to the Langmuir isotherm and the pseudo-second-order kinetic model. The results of the FT-IR analysis demonstrated strong agreement with the pseudo-second-order chemisorption process description. The maximum adsorption capacity was calculated from the Langmuir isotherm and evaluated to be 166.66 mg.g^−1^. The positive ∆H (12.19 J.mol^−1^) indicates that the adsorption process is endothermic. Almond shell was found to be a stable adsorbent. Three different statistical design sets of experiments were taken out to determine the best conditions for the batch adsorption process. The optimal conditions for MG uptake were found to be adsorbent mass (m = 0.1 g), initial dye concentration (C_0_ = 600 mg.L^−1^), and temperature (T = 25 °C). The analysis using the D-optimal design showed that the model obtained was important and significant, with an R^2^ of 0.998.

## 1. Introduction

The existence of dyes in water affects aquatic life due to their carcinogenic and mutagenic properties [1]. Therefore, it is necessary to effectively remove dyes from water. Treatment technologies such as coagulation [2], photocatalytic degradation technique [3], filtration through membranes [4], and adsorption [5,6,7,8,9,10] are used. The adsorption technique is one of the most cost-effective, efficient, and eco-friendly methods, as no byproducts are formed during the purification process. The development of new effective and economical natural materials that can be applied on a large scale has attracted scientific interest.

Activated carbon (AC), the most commonly used adsorbent, is excellent at removing a wide range of organic contaminants. In order to reduce the amount of colorants released into the environment, activated carbon adsorption is widely used in industrialized countries to remove dyes from water and wastewater [6]. Many economic and industrial fields, such as electronics, cosmetics, energy, petroleum, mining, pharmaceutical, chemical, automotive, vacuum manufacturing, and nuclear energy industries, use this type of materials in their manufacturing infrastructure [11]. Most of the above applications require carbons with specific chemical properties on their surfaces [12]. The natural materials are applied to adsorb dyes, treat odors, and remove heavy metals. According to previous research, several types of lignocellulosic agricultural waste have been tested for the adsorption removal of different pollutants [10,11,12,13,14]. To only cite a few, date pits activated carbon [15], date palm biochar [16], nut shell-derived carbon [17], pistachio shells [18], edamame shell [19], peanut shell [20], etc. In addition, one of the most abundant lignocellulosic agricultural waste products is almond shell (AS). It served as the raw material used to produce AC. Various applications of activated carbon-based AS have been studied [10,11,14]. This material could be used in agricultural operations [21] to remove Direct Red 80 dye [22] and to eliminate Orange G dye and hexavalent chromium ions from aqueous matrices [23]. Moreover, the adsorption study of AS for bromophenol blue (BPB) [24] and for methyl orange (MO) [25] was investigated. It has also been used to adsorb lead and cadmium from aqueous solution [26]. In order to remove Cu(II) from water, a batch adsorption study was reported by the adsorption process using AS as the adsorbent [27]. The characteristics of AS are not well understood. Xuemin et al. showed the physical structure and basic chemical composition of AS in order to investigate these biomass properties [28].

Malachite green (MG) is a cationic, highly toxic dye that is well suited for dyeing cotton, leather, and dyeing wool. It is also widely used in aquaculture and animal husbandry as a fungicide and disinfectant. Despite the diverse applications of MG, it has been found to have mutagenic and carcinogenic properties and also leads to several types of diseases [29]. Numerous studies have been carried out to investigate the removal of MG using an adsorption process [30]. Some of these articles are based on the use of commercial activated carbons [31,32], and others are based on activated carbons prepared from agricultural wastes. For example, Ahmad et al. [33] prepared activated carbon from durian seeds and studied the removal of MG; Akar et al. [34] tried to use tea leaves to eliminate MG from water; Basirun et al. [35] prepared activated carbon from the banana stalk as a lignocellulosic agricultural waste. MG has also been removed from an aqueous solution using ceramic clays as adsorbents [36].

Most of the adsorption processes reported in the literature have used activated carbon as the adsorbent. This choice is because activated carbon provides a high surface area, so the amount of contaminant that can be adsorbed is large. Carbon’s surface area limits the kind of pollutants it can absorb, but the majority of its pores are micropores, which allow it to effectively adsorb small molecules [37]. On the other hand, the almond shell is considered a waste that leads to environmental issues. AS waste has approximately no significance in an industrial application, so it is usually disposed of in landfills or burned in the open air. Thus, making almond waste useful will boost almond production and reduce environmental risks from its byproducts. Likewise, it successfully realized the recycling of agricultural waste and reduced its pollution to the environment. Although the use of carbon is very limited, it can avoid a higher relative adsorption capacity than some activated carbons. In this work, carbon prepared from almond shell (AS), kinetic and thermodynamic study was discussed. The main aim of the current study is to determine the relationship between adsorption capacity and affecting parameters; the design of the experiment is used to optimize the experimental conditions. The adsorption process evaluation parameters have been provided. In addition, the effects of different parameters were investigated using a D-optimal design to obtain the optimal factors leading to the adsorption process, making it a significant future contribution to the adsorption processing.

## 2. Results and Discussion

### 2.1. Adsorbent Characterization

The FTIR spectrum of the sorbent carbon before adsorption shows functional groups present on the surface. Figure 1a shows a peak at 3360 cm^−1^, which may represent the stretching vibration (O-H) of alcohols and phenols or the vibration of (N-H) groups [24,38]. The peak at 1704 cm^−1^ indicates the presence of a carbonyl group (C=O) corresponding to ketones and carboxylic groups [34,39]. The characteristic peak at 1580 cm^−1^ indicates that the carbon of the almond shell contains more lignin [28]. The band at 1592 cm^−1^ is due to conjugated C=C [24]. The band at 1178 cm^−1^ belongs to (C-O) vibration; the peak at 993 cm^−1^ might be due to (C-H) in aromatics structure containing two adjacent hydrogens per ring and isolated aromatic hydrogens [24]. A peak that appeared at 751 cm^−1^ may correspond to out-of-plane vibrations of the (C-H) deformation. After MG adsorption, the observed peaks were shifted (1000 to 993 cm^−1^, 1588 to 1580 cm^−1^, 1689 to 1704 cm^−1^, and 3378 to 3360 cm^−1^). This clearly indicates that the adsorption process is accompanied by the formation of new bonds between the MG and the almond shell. The results here are consistent with the pseudo-second-order description of the chemisorption process. 

The XRD pattern of the AS sample and after the adsorption of MG is shown in Figure 1b. The AS shows two broad peaks in the 2θ range of 23.82° and 43.00°, corresponding to the (002) and (110) crystal planes of graphite materials (JCPDS-ICDD 75–1261), resulting in a good layer orientation, but the absence of a strong peak indicates that the structure is largely amorphous. Similar results were found in the scientific literature for the production of activated carbon from AS using carbon dioxide activation [38,40]. The X-ray diffraction reflection peaks found in the activated carbons are due to the regular deposition of carbon [40] or to the metal oxides found in the coal structure [40]. It can be identified by comparing the peaks found with the pattern of XRD with the list of diffraction database files. In addition, the amorphous structure is an advantage in adsorbent materials. Moreover, the intense peaks show the presence of a highly organized crystalline structure of AS after the adsorption of MG; the intensity of some peaks is significantly decreased in the AS loaded with MG. This was attributed to the adsorption of MG on the upper layer of the crystalline structure of the AS surface.

The AS degrades in two steps (see Figure 2). The first mass loss (2.82 wt %) between 30 °C and 120 °C is due to the removal of moisture from the structure. The second mass loss (21.26 wt %) between 190 and 450 °C can be attributed to the degradation of cellulose, hemicellulose, and lignin in the structure and the removal of gaseous products and volatile components from the structure during decomposition [41] and the total mass loss is 24.92 wt %.

The microstructure of carbon-based almond shell presents an amorphous structure that gives a highly porous surface; this result showed a good agreement with textural characterization obtained by the BET method. The structure of the main parameters is shown in Table 1. The nitrogen adsorption/desorption isotherm of the prepared adsorbent is shown in Figure 3a. Based on the isotherm shape, it can be determined that the isotherms are of type H4 [42]. Micropores are shown in the presence of mesopores in the t-plot from type H4 isotherm in Figure 3a. The external surface area can be achieved based on the straight line’s slope to smaller t values (t < 0.5) [43]. In this range, adsorbed molecules in large pores are taken up. The micropore surface area can be calculated by subtraction of the outer area from the BET surface area. For larger values of t (0.5 < t < 0.8), the micropore volume is the Y-intercept of the second straight line.

The morphology of the AS sample was revealed by SEM and is shown in Figure 3b. The image showed that the external surface of AS was full of cavities and quite irregular because of the milling treatment. The AS was subjected to ball milling and heating to 400 °C, the particles became irregular and crushed, and many small cavities appeared across the surface. Because of these well-developed pores, the AS possesses a high surface area.

### 2.2. Batch Adsorption Studies

Figure 4a illustrates the way adsorbent mass affects the removal of MG from AS. The results show that the adsorption capacity of MG decreases as the amount of adsorbent increases. Since the adsorption capacity is determined per unit of adsorbent mass, the decreasing adsorption capacity can be explained by the unit of adsorbent (Equation (1)).

The initial dye concentration of MG on 0.1 g of AS was studied from 5 to 600 mg.L^−1^. As shown in Figure 4b, the adsorption capacity is increased from 1.22 to 166.66 mg.g^−1^. It was observed that all the MG present in the medium is adsorbed on the adsorbent surface, indicating that the adsorbent is rich in adsorption sites. The increasing amount of adsorption indicates the efficiency of the adsorption of MG on AS and indicates that this AC presents a high adsorption capacity for a wide concentration range, which is very important in industrial applications.

The impact of contact time was generally studied at different concentrations (50, 100, 300, and 500 mg.L^−1^), and we observed that AS was found to be a stable adsorbent, and many researchers obtained the same results for dyes at different concentration [22,24].

The adsorption of MG dye (50 mg.L^−1^) on AS is shown in Figure 4c; the adsorption was studied at 25 °C for 5 to 35 min. The results show a rapid removal of the dye on activated carbon, which is an advantage for this adsorbent. A rapid removal of MG was observed in 5 min, and the amount adsorbed did not change significantly until 35 min due to saturation of adsorption sites on the AS.

Figure 5a,b show the Langmuir and Freundlich adsorption isotherms of the AS adsorbent by linear analysis. Table 2 summarizes the corresponding isotherm parameters. According to the R^2^, the Langmuir model fitted the experimental data best by linear analysis, while the Freundlich model fitted the worst. The shapes of these adsorption models indicate that the adsorption is monolayer with a higher correlation coefficient R^2^ equal to 0.956 higher than the R^2^ value of Freundlich, and the maximum adsorption capacity is 166.66 mg.L^−1^.

A comparison with other systems used for MG adsorption indicated that the AS adsorbent possessed a high maximum adsorption capacity, indicating a higher affinity between AS and MG than for other systems [44,45,46,47,48,49,50,51,52,53] (Table 3).

### 2.3. Thermodynamic Investigations

The effect of temperature on the adsorption of MG on AS has been studied, and it is necessary to evaluate the thermodynamic parameters. At a dye concentration of 50 mg.L^−1^, batch studies with 0.1 g of AS were useful. The adsorption was carried out at 25 °C, 50 °C, 75 °C and 100 °C. The following formulas were used to calculate the free energy change (ΔG), enthalpy change (ΔH), and entropy change (ΔS), which are the thermodynamic parameters [9,10]:(1)$${\mathrm{lnK}}_{D}=\frac{\Delta S}{R}-\frac{\Delta H}{\mathrm{RT}}=-\frac{\Delta G}{\mathrm{RT}}$$
(2)ΔG=ΔH−TΔS
(3)$$K_{D}=\frac{q_{e}}{C_{e}}$$
where:

R: the gas constant (8.314 J.mol^−1^.K^−1^).

K_d_: the distribution coefficient.

T: the temperature (K).

q_e_: equilibrium adsorption capacity (mg.g^−1^).

C_e_: residual dye concentration (mg.L^−1^).

Experimental data show a slight increase in adsorption capacity in response to an increase in temperature (Figure 6). It is clear from the negative ∆G that this adsorption process is spontaneous [33,42]. The ΔH and ΔS were estimated from the slope and intercept of this plot. ΔH and ΔS values were 12.19 J.mol^−1^ and 20.24 J.mol^−1^K^−1,^ as listed in Table 4. Moreover, the endothermic nature of the adsorption process is confirmed by the positive ∆H [54].

The positive value of ∆S presents the affinity between MG and AS and indicates an increase in the level of dye species randomness.

### 2.4. Adsorption Kinetic Study

The models of intraparticle diffusion, pseudo-first-order, and pseudo-second-order were used to investigate the kinetics of MG adsorption on AS. The pseudo-first-order can be expressed as:(4)$$\ln(q_{e}-q_{t})=\ln q_{e}-k_{1}t$$
where:

q_e_: theoretical adsorption capacity at equilibrium.

q_t_: the adsorption capacity at time t.

K_1_: pseudo-first-order rate constant.

The adsorption is illustrated by the pseudo-second-order model describing the chemisorption process, which is written as:(5)$$\frac{t}{q_{t}}=\frac{1}{k_{2}q_{e}^{2}}+\frac{1}{q_{e}}t$$
where:

K_2_: pseudo-second-order rate constant.

The following equation provides the intraparticle diffusion model as:(6)$$q_{t}=K_{\mathrm{id}}t^{1/2}+C$$
where:

K_id_: the rate constant for the intraparticle diffusion model.

A representative model is considered to be the best fit based on the R^2^ value. According to Table 5, the pseudo-second-order model has the highest correlation coefficient (R^2^ = 0.999) (Figure 6b), surpassing both the intraparticle diffusion model (R^2^ = 0.832) and the pseudo-first-order model (R^2^ = 0.759). It is clear that the pseudo-second-order model was the best one to use to explain the adsorption process. Moreover, Figure 4c displays that the system is very close to the equilibrium within 5 min. Therefore, according to the study by Simonin, caution should be exercised when comparing PFO kinetics with PSO kinetics [55]. Because for the PFO model, when q_t_ approaches q_e_, the value of (q_e_ – q_t_) becomes smaller and smaller, leading to the accuracy reduction of k1. In addition, taking much data at equilibrium into account in the PSO kinetic model study is not coherent. As the fitting of the intraparticle diffusion kinetic model does not involve the data at equilibrium, and the coefficient R^2^ of the intraparticle diffusion model is 0.832, the intraparticle diffusion is predominant for MG adsorption, and the process was controlled by diffusion.

### 2.5. D-Optimal Designs for 3 Parameters

The models determine the impact of experimental conditions and to examine the individual effects of the adsorbent content, a D-optimal design was employed, starting dye concentration and temperature as well as their double interactions.

The mode program was used to examine the outcomes. Equation (10) describes the coded model used for the D-optimal design.
(7)$$Y_{i}=\beta_{0}+\beta_{1}X_{1i}+\beta_{2}X_{2i}+\beta_{3}X_{3i}+\beta_{12}X_{1i}X_{2i}+\beta_{13}X_{1i}X_{3i}+\beta_{23}X_{2i}X_{3i}$$

$Y_{i}$ is the expected response; $X_{\mathrm{ji}}$ are values (j = 1, 2, 3; i = 1, 2, 3… 9), representing the respective parameters in their coded forms; $\beta_{0}$ is the average response value; and $\beta_{1}$, $\beta_{2}$, and $\beta_{3}$ are the linear coefficients. The coefficients $\beta_{1}$, $\beta_{2}$, and $\beta_{3}$ represent the effects of adsorbent mass (m), initial adsorbate concentration (C_0_), and temperature (T), respectively (Table 6).

The main effects and two-way interactions were found to be statistically significant, with the *p*-value being at the lowest level [56]. The main effects (Table 6) illustrate the average variances between low and high levels. In this case, the factor that has a positive effect, as it increases from low to high levels, causes an increase in adsorption capacity. In general, high coefficients of determination (>0.99) were obtained for all dependent variables. The following model equation was obtained based on the level of parameters:(8)$$q_{e}=37.4-8.2m+40.7C_{0}-5.9T-21.8{m*C}_{0}+24.8m*T-22.8C_{0}*T$$

As can be seen in Figure 7, the model perfectly represented the results of the experiment. When a factor has a negative sign, it means that the result improves as the value of the parameter decreases. The results showed that a high level of adsorption capacity resulted in a higher initial concentration of dye with a significant effect of this factor. Thus, increasing the adsorbent content and temperature decreases the response. In addition, significant interactions between the factors were observed, which means that the individual effect of each factor is related to the levels of each factor controlling the response. Table 7 summarizes the outcomes of the optimization to obtain the highest possible value of the adsorption capacity. The obtained predicted results (Figure 7) are in close agreement with the actual experiment, which is proven by the adjusted R^2^ (0.998), including the double interaction.

The following settings were found to be ideal for MG uptake: m = 0.1 g of adsorbent mass, C_0_ = 600 mg.L^−1^ of initial dye concentration, and T = 25 °C of temperature.

## 3. Materials and Methods

### 3.1. Materials

Hydrochloric acid (HCl) (Merck reagent grade, 37%, Darmstadt, Germany), sodium hydroxide (NaOH) (Merck, Darmstadt, Germany), acetone (Sigma Aldrich, 90%, Taufkirchen, Germany), and deionized water. The malachite green oxalate has chemical formula C_46_H_50_N_4_.3C_2_H_2_O_4_; M = 929.02 g.mol^−1^ (Sigma Aldrich, Taufkirchen, Germany), and maximum absorbance λ_max_ = 617 nm. A total of 1 g.L^−1^ of MG was prepared in distilled water to be used in the adsorption study.

### 3.2. Physical Modification of the Natural Material

The almond shells were produced on a local farm in Mascara (Algeria). The biomass was ground and sieved before being washed with distilled water for 24 h to eliminate dust from its surfaces. The material was then filtered, cleaned with acetone and distilled water to remove organic contaminants, and dried in an oven at 60 °C for 24 h. Afterwards, the material was heated to 400 °C in an air atmosphere for 75 min [57]. Then, a graphite container was used to hold the carbon adsorbent [58].

### 3.3. Characterization of the Adsorbent

The surface properties of the prepared carbon were determined by FTIR spectroscopy using a Bruker Alpha spectrophotometer. The almond shell was analyzed by X-ray diffraction (Goniometer/MiniFlex 300/600-Diffracted beam mono/Bent-Detector/SC-70, Tokyo, Japan) with an X-ray source (Cu Kα) operating at 15 mA and 40 kV. X-ray diffraction was performed in the range of 2θ = 1.5° with a step width of 0.03° and a step duration of 1 s. The texture of AC was characterized at 77 K (Quantachrome/Autosorb-6, Madrid, Spain). The samples were outgassed at 300 °C under vacuum for 2 h. The nitrogen adsorption results were used to calculate the BET surface area. Thermograms of the samples were performed between 25 °C and 700 °C under an N_2_ atmosphere at heating rates of 10 °C per minute using a PerkinElmer Diamond TGA device (Washington, DC, USA).

### 3.4. Investigation of Parameters via Adsorption Procedure

In order to evaluate the amount of MG adsorbed, 0.1 g of raw almond shell material was added to 25 mL (50 mg.L^−1^) of MG at ambient temperature (25 °C) under magnetic stirring at pH = 5 [39]. The variations of absorbance were measured by UV-vis (Kyoto, Japan) at a maximum wavelength (λ_max_ = 617 nm) at different time intervals (5, 10, 15, 20, 25, 30, and 35 min) during the adsorption process in order to study the effect of contact time. The effect of initial dye concentration was studied from 5 to 600 mg.L^−1^ for 35 min. The effect of the amount of adsorbent was studied at different amounts of AS (0.1 g to 0.6 g) in 50 mg.L^−1^ of dye solution. The adsorption amount (q) was given by the following expression:(9)$$q=\frac{(C_{0}-C_{t})V}{m}$$

C_0_ and C_t_ are the concentration levels of initial and residual colorant (mg.L^−1^), while V (L) is dye volume and m is the amount of carbon adsorbent (g). 

### 3.5. Adsorption Isotherms

In order to better understand the relationship between the adsorption capacity and MG dye concentration, Freundlich and Langmuir isotherms were used. These two isotherms were chosen because they can be used for a wide range of adsorbate concentrations. The linearized form of the Langmuir model [44]:(10)$$\frac{C_{e}}{q_{e}}=\frac{1}{K_{l}q_{m}}+\frac{C_{e}}{q_{m}}$$

The linearized equation of Freundlich [58]:(11)$${\mathrm{lnq}}_{e}={\mathrm{lnK}}_{f}+\frac{1}{n}{\mathrm{lnC}}_{e}$$
where:

q_e_: equilibrium capacity of adsorption (mg.g^−1^).

C_e_: concentration of residual dye (mg.L^−1^).

q_m_: the maximum capacity of adsorption to form a monolayer (mg.g^−1^).

K_L_: Langmuir equilibrium constant (L.mg^−1^).

n: Freundlich constant presents the bond energies between the dye and adsorbent.

K_F_: Freundlich equilibrium constant (L.mg^−1^).

### 3.6. D-Optimal Design of Experiments

D-optimal designs are based on an exchange system that uses computers to generate the best possible set of experiments.

A D-optimal design was used to study the impact of different parameters: the temperature during adsorption (T), the initial concentration of the dye (C_0_), and the mass of the adsorbent (m). The most significant factor influencing the adsorption process was identified. D-optimal designs are based on the information matrix to predict and optimize the experimental conditions for a particular model. The real values and the response, the adsorption capacity (Y), are shown in Table 8.

For any process, it is important to know the influence of different physicochemical parameters (also termed control factors) on the results of the process. Factorial design is used to reduce the total number of experiments in order to achieve the best percentage removal (%MG). The factorial design determines which factors have important effects on a response (%MG) as well as how the effect of one factor varies with the level of the other factors. The number of experimental runs at two levels is 2^k^, where k is the number of factors. Today, the most widely used kind of experimental design to estimate main effects as well as interaction effects is the 2^k^ factorial design in which each variable is investigated at two levels. The three factors considered were the adsorbent mass, initial MG concentration, and temperature. The high and low levels represented by +1 and −1, respectively, defined for the 2^3^ factorial designs, are listed in Table 9. The low and high levels for the factors were selected according to preliminary experiments.

## 4. Conclusions

It was observed that almond shells could be an effective adsorbent for malachite green by a simple method of preparing a carbon material. The results show a good adsorption capacity, 166.66 mg.g^−1^, obtained for 600 mg.L^−1^. It was found that the almond shell can be used in a wide concentration and temperature range. Experimental data of malachite green adsorption onto AS followed the Langmuir isotherm. The kinetic study shows that the adsorption could be well controlled by the pseudo-second-order model. The ideal conditions (adsorbent content, initial dye concentration, and temperature) for MG adsorption on AS were determined using a D-optimal design. Increasing the concentration from 5 to 600 mg.L^−1^ promoted the adsorption. The following parameters were found to be optimal for MG uptake: m = 0.1 g of adsorbent mass, C_0_ = 600 mg.L^−1^ of initial color concentration, and T = 25 °C temperature. At the 95% confidence level, all variables and interactions of the experimental design were statistically significant.

## Figures and Tables

**Figure 1 molecules-29-00054-f001:**
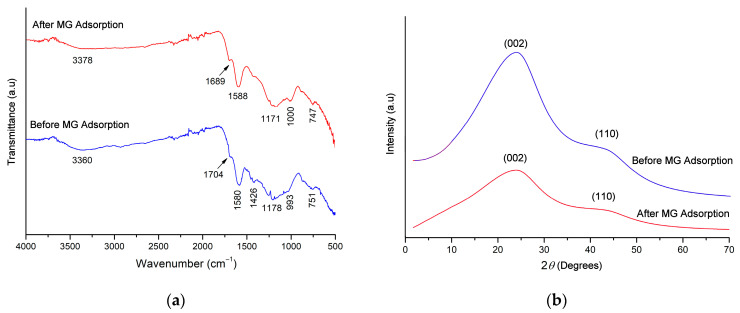
(**a**) Infrared spectra of almond shell carbon and (**b**) X-ray diffraction of almond shell carbon.

**Figure 2 molecules-29-00054-f002:**
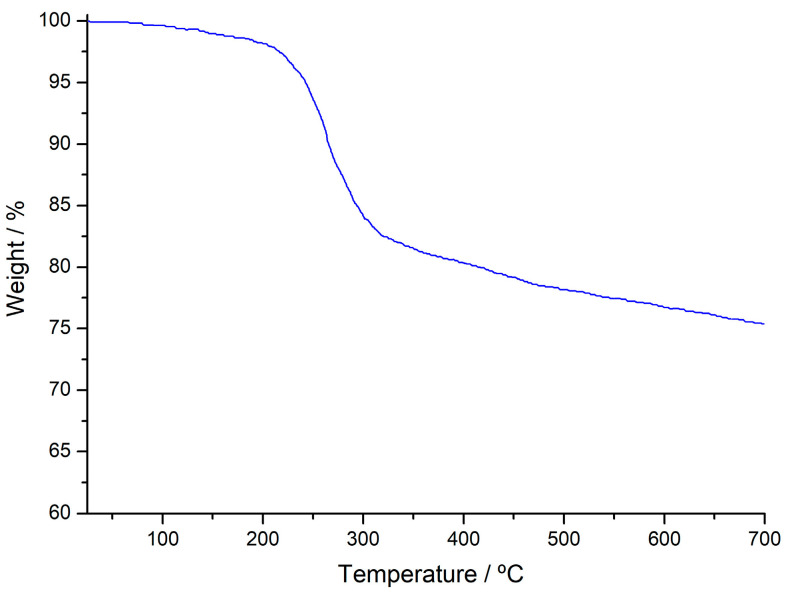
Thermogravimetry (TG) curve of AS sample.

**Figure 3 molecules-29-00054-f003:**
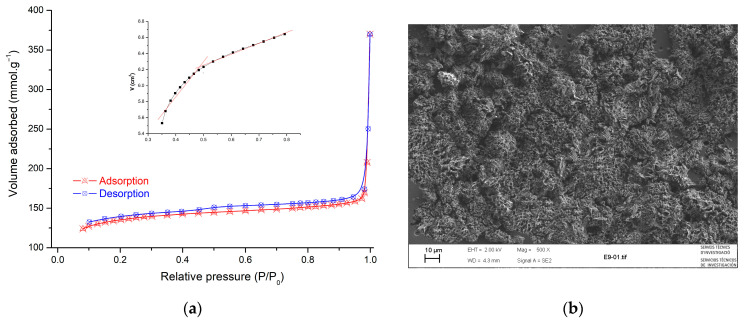
(**a**) Nitrogen adsorption/desorption isotherms at 77 K and t-plot fitting and (**b**) SEM image of AS sample.

**Figure 4 molecules-29-00054-f004:**
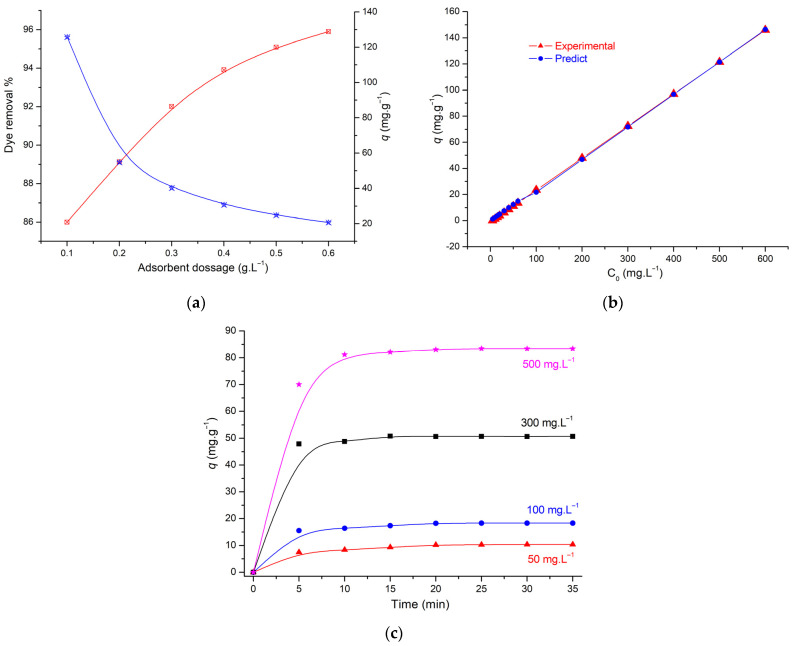
(**a**) Effect of amount of adsorbent (0.1 g, 25 °C, pH = 5.0; C_0_: 50 mg.L^−1^), (**b**) experimental and predicted effect of initial concentration dye, and (**c**) effect of contact time (0.1 g, 25 °C, pH = 5.0).

**Figure 5 molecules-29-00054-f005:**
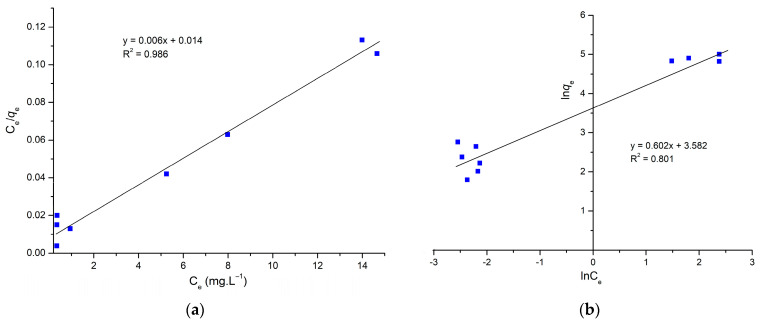
(**a**) Linearized form of Langmuir and (**b**) linearized form of Freundlich.

**Figure 6 molecules-29-00054-f006:**
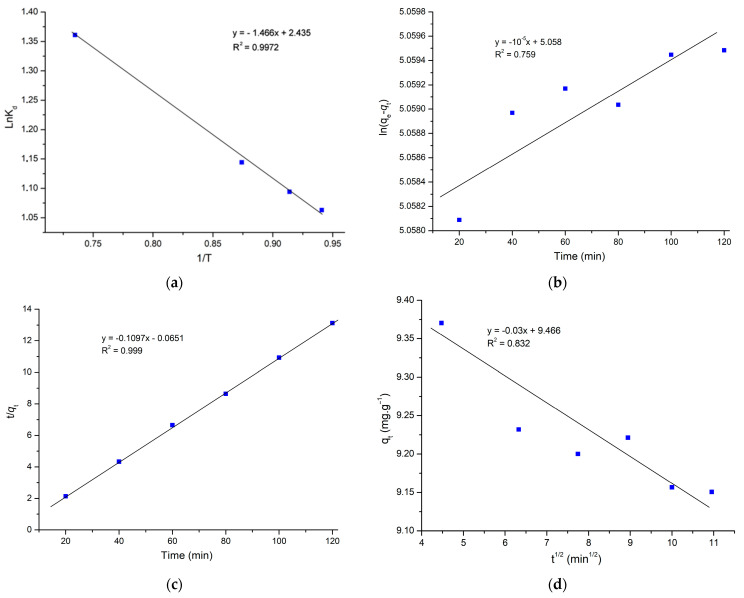
(**a**) Plot ln_kD_ Vs 1/T, (**b**) pseudo-first-order model, (**c**) pseudo-second-order model, and (**d**) intraparticle diffusion kinetic model (C_0_ = 50 mg.L^−1^, 0.1 g, 25 °C).

**Figure 7 molecules-29-00054-f007:**
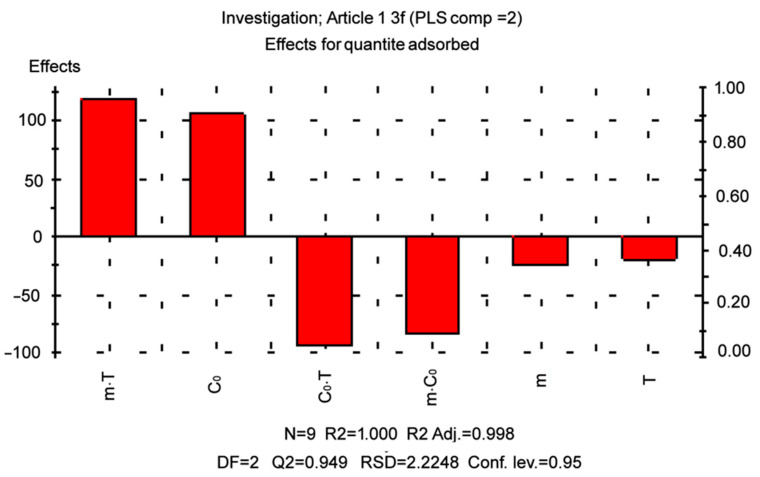
Effects of adsorbent content, initial dye concentration, and temperature on the adsorption capacity.

**Table 1 molecules-29-00054-t001:** Porous structure parameters of almond shell.

Parameters	
BET surface area (m^2^.g^−1^)	120.21
Pore volume (cm^3^.g^−1^)	0.572
Pore diameter (nm)	9.263
Micropore volume (cm^3^.g^−1^)	0.213

**Table 2 molecules-29-00054-t002:** Isotherm parameters for the adsorption of MG onto AS obtained from equilibrium models.

Isotherm Model	Parameters	Values
Langmuir	q_m_	166.66 (mg.g^−1^)
	K_L_	428 (L.g^−1^)
	R^2^	0.986
	R_L_	0.0034
Freundlich	n	1.661
	K_F_	35.945 (mg.g^−1^)
	R^2^	0.801

**Table 3 molecules-29-00054-t003:** Maximum adsorption capacities of MG on various adsorbents.

Adsorbents	q_m_ (mg.g^−1^)	pH	C_0_ (mg.L^−1^)	References
Tunisian almond shell	126.90	6.0	200	[44]
Almond shell (*P. dulcis*)	22.30	//	600	[45]
Walnut shell	90.80	3.8	400	[46]
Avena sativa (oat) hull	51.42	8.0	200	[47]
Polydopamine–chitosan nanoparticles	60.97	8.0	300	[48]
Rattan sawdust	62.71	10.0	300	[49]
Apricot stone (ASAC)	23.80	10.0	10	[50]
Bamboo leaves	98.00	6.0	100	[51]
Coffee bean	16.07	4.0	100	[52]
Wood apple shell	34.56	7.5	100	[53]
Almond shell treatment at 400 °C	166.66	5.0	600	This study

**Table 4 molecules-29-00054-t004:** Thermodynamic parameter values of MG adsorption on AS.

∆G (KJ.mol^−1^)				∆S (J.mol^−1^K^−1^)	∆H (J.mol^−1^)
298 K	323 K	348 K	373 K	20.24	12.19
−0.76	−0.16	−0.26	−0.41		

**Table 5 molecules-29-00054-t005:** Parameter values for different kinetic models at 25 °C and C_0_ = 50 mg.L^−1^.

Model	Parameter	Value
Pseudo-first-order	K_1_ (min^−1^)	−10^−5^
	q_e_ (mg.g^−1^)	157.27
	R^2^	0.759
Pseudo-second-order	q_e_ (mg.g^−1^)	9.17
	K_2_ (g.mg^−1^.min^−1^)	0.183
	R^2^	0.999
Intraparticle diffusion	K_id_ (mg.g^−1^.min^½^)	−0.03
	C (mg.g^−1^)	9.466
	R^2^	0.832

**Table 6 molecules-29-00054-t006:** Statistical parameters with the use of D-optimal design.

Term	Effect	Coef	SE Coef	P
Without interactions				
Constant		43.5778	0.4579	2.4290 × 10^−9^
m	−6.6792	−2.2264	0.5064	0.0070
C_0_	142.609	55.0073	0.5146	1.3589 × 10^−9^
T	−1.6715	−0.5259	0.5042	0.3447
R^2^	0.999			
DF	5			
2-way interactions				
Constant		37.3984		
m	−24.720	−8.2400		
C_0_	105.606	40.7344		
T	−18.838	−5.9278		
M × C_0_	−84.981	−21.852		
M × T	118.225	24.8007		
C_0_ × T	−93.860	−22.784		
R^2^	0.998			
DF	2			

**Table 7 molecules-29-00054-t007:** Optimization for MG adsorption on AS using statistical design.

m	C_0_	T	Y_i_
0.2985	20.0045	70.8051	137.333
0.1	600	25	158.015
0.2399	20.0245	100	167.043
0.3	108.771	89.7654	173.242
0.3	20.0001	88.75	197.009
0.1	599.994	34.2698	135.901
0.1	599.99	25.0117	157.982
0.1	600	30	146.089

**Table 8 molecules-29-00054-t008:** Matrix for experimental design of MG adsorption on almond shell carbon.

Experiments	m	C_0_	T	Y
1	0.3	50	30	3.99
2	0.1	50	25	9.95
3	0.1	50	100	9.48
4	0.1	20	30	4.96
5	0.1	60	30	14.98
6	0.1	300	30	71.75
7	0.1	500	30	121.49
8	0.1	600	30	146.45
9	0.1	50	30	9.15

**Table 9 molecules-29-00054-t009:** The levels of the experimental factors.

Factors	Low Factor Level	High Factor Level
	−1	+1
Adsorbent mass, m (g)	0.1	0.3
Initial concentration of the colorant, C_0_ (mg.L^−1^)	50	600
Temperature during adsorption, T (°C)	25	100

## Data Availability

All data obtained in this study are part of this paper.

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
