# Peer review of "Optimization Conditions of Malachite Green Adsorption onto Almond Shell Carbon Waste Using Process Design"

_molecules, 2023, doi:10.3390/molecules29010054_

Round 1

Reviewer 1 Report (Previous Reviewer 1)

Comments and Suggestions for Authors

In this manuscript, the author reported the preparation of almond shell based biocarbon and the removal of Malachite green. The whole manuscript was in bad organizing and writing, major revision was needed.

(1) Extensive editing of English language required. Eg. Line 17, it should be Almond shell based biocarbon is a cheap adsorbent for the the removal of Malachite green, which has been investigated in this work. Line 18, modified almond shell should be almond shell based biocarbon.....

(2) As mentioned in the introduction, there are many reports about biocarbon for the removal of MG, what is the novelty of this work?

(3) Except for MG, grade and supplier of other chemicals used in this work should also present in detail.

(4) Table 2 was not cited in the manuscript.

(5) The SEM images of the obtained almond shell based biocarbon should provided.

(6) Figure 4b should moved to the section of 3.2. In addition, the caption of X-axis should be adsorbent amount, rather than Relative pressure.

(7) For Table 5, to compare the maximum capacity, the initial concentration of MG should also provided, as for Ref. 39, the maximum concentration is 200 mg/L, the qm is 126.9 mg/g. However, maximum capacity of 166.66 mg/g was obtained at 600 mg/g for this work, which demonstrated that the adsorption efficiency of AS obtained in this is not so high.

(8) The kinetic analysis concludes that the experimental data fit to a pseudo second order kinetics (Table 7). However, as highlighted by Simonin (http://dx.doi.org/10.1016/j.cej.2016.04.079), caution should be exercised when comparing pseudo first order kinetics with pseudo second order kinetics and data close to the equilibrium are included in analysis, as seems to be the case for this work, and according to Fig 5b, within 5 min the system is very close to the equilibrium. I strongly recommend the reading of the paper by Simonin and Gao (Polymers 2023, 15, 4048) to discard bias in the fitting and discuss it in the revised version of the manuscript.

Comments on the Quality of English Language

 Extensive editing of English language required

Author Response

Response to Reviewer 1 Comments

Point 1: Extensive editing of English language required. Eg. Line 17, it should be “Almond shell based biocarbon is a cheap adsorbent for the the removal of Malachite green, which has been investigated in this work.” Line 18, “modified almond shell” should be “almond shell based biocarbon”.....

Response 1: We have done extensive editing for the English language. We have rewritten the following sentences (see line 17 and line 19)

“Almond shell based biocarbon is a cheap adsorbent for the the removal of Malachite green, which has been investigated in this work.”

And

“almond shell based biocarbon.”

Point 2: As mentioned in the introduction, there are many reports about biocarbon for the removal of MG, what is the novelty of this work?.

Response 2: What is new in this work is the recycling of agricultural waste, and for this we have added the following sentence (see lines 83-89)

“In the other hand, almond shell is considered a waste that leads to environmental issues. AS waste has approximately no significance in an industrial application, so usually disposed of in landfills or burned in the open air. Thus, making almond waste useful will boost almond production and reduce environmental risks from its byproducts. Likewise, it successfully realized the recycling of agricultural waste and reduced its pollution to the environment.”

 Point 3: Except for MG, grade and supplier of other chemicals used in this work should also present in detail.

Response 3: We presented the grade and supplier of chemicals used in this work (see 2.1. Material section).

Point 4: Table 2 was not cited in the manuscript.

Response 4: We added this paragraph related to Table 2 (see lines 158-169)

“For any process, it is important to know the influence of different physicochemical parameters (also termed control factors) upon the results of the process. Factorial design is used to reduce the total number of experiments in order to achieve the best percentage removal (%MG). The factorial design determines which factors have important effects on a response (%MG) as well as how the effect of one factor varies with the level of the other factors. The number of experimental runs at two levels is 2k, where k is the number of factors. Today, the most widely used kind of experimental design, to estimate main effects as well as interaction effects, is the 2k factorial design in which each variable is investigated at two levels. The three factors considered were the adsorbent mass, initial MG concentration and temperature. The high and low levels represented by +1 and -1, respectively defined for the 23 factorial designs were listed in Table 2. The low and high levels for the factors were selected according to preliminary experiments.”

Point 5: The SEM images of the obtained almond shell based biocarbon should provided..

Response 5: We added Figure 3b of SEM image with the following paragraph (see lines 227-232).

“The morphology of the AS sample were revealed by SEM and are shown in Figure 3b. The image showed that the external surface of AS was full of cavities and quite irregular because of the milling treatment. The AS was subjected to ball milling and heating to 400°C, the particles became irregular and crushed, and many small cavities appeared across the surface. Because of these well-developed pores, the AS possesses a high surface area”.

Point 6: Figure 4b should moved to the section of 3.2. In addition, the caption of X-axis should be adsorbent amount, rather than Relative pressure.

Response 6: We moved Figure 4a to Section 3.2. In addition, we corrected the x-axis in Figure 4a

Point 7: For Table 5, to compare the maximum capacity, the initial concentration of MG should also provided, as for Ref. 39, the maximum concentration is 200 mg/L, the qm is 126.9 mg/g. However, maximum capacity of 166.66 mg/g was obtained at 600 mg/g for this work, which demonstrated that the adsorption efficiency of AS obtained in this is not so high.

Response 7: We added the initial concentration of MG with the pH values to compare the maximum capacity.

Point 8: The kinetic analysis concludes that the experimental data fit to a pseudo second order kinetics (Table 7). However, as highlighted by Simonin (http://dx.doi.org/10.1016/j.cej.2016.04.079), caution should be exercised when comparing pseudo first order kinetics with pseudo second order kinetics and data close to the equilibrium are included in analysis, as seems to be the case for this work, and according to Fig 5b, within 5 min the system is very close to the equilibrium. I strongly recommend the reading of the paper by Simonin and Gao (Polymers 2023, 15, 4048) to discard bias in the fitting and discuss it in the revised version of the manuscript.

Response 8: We added the following paragraph (see lines 320-328)

“Moreover, the Figure 4c display that the system is very close to the equilibrium within 5 min. Thereby, according to the study of Simonin, it caution should be exercised when comparing PFO kinetics with PSO kinetics [57]. Because for the PFO model, when qt approaches qe, the value of (qe–qt) becomes smaller and smaller, leading to the accuracy reduction of k1. In addition, taking much data at equilibrium into account in the PSO kinetic model study is not coherent. As the fitting of Intraparticle diffusion kinetic model do not involve the data at equilibrium, and the coefficient R2 of the Intraparticle diffusion model is 0.832, the Intraparticle diffusion is predominant for MG adsorption, and the process was controlled by diffusion”.

[57] Miao, P.; Sang, Y.; Gao, J.; Han, X.; Zhao, Y.; Chen, T. Adsorption and Recognition Property of Tyrosine Molecularly Imprinted Polymer Prepared via Electron Beam Irradiation. Polymers. 2023, 15, 4048. https://doi.org/10.3390/polym15204048

Reviewer 2 Report (Previous Reviewer 2)

Comments and Suggestions for Authors

Authors have resubmitted the manuscript, seems few concerns were not taken up in the revised manuscript,

1. Introduction should focus on the adsorbent materials which should include different kind of adsorbents prepared form the same precursor materials, types of pollutants been studies, method of adsorbent synthesis etc.

2. Authors were asked to check the references but seems it was not checked properly, reference 13 needs to be cross checked for citing reference for agricultural waste, this article deals in conducting polyaniline as adsorbent.

3. Similarly refrence 19 in the article in Yusop, M.F.M. and co-worker not Ahmed et al.,

4. Akar et al is not at reference number 30. The actual authors of the mentioned title are Mohd Azmier Ahmad, Norhidayah Ahmad & Olugbenga Solomon Bello

5. Table 5 needs to be enrich, at least it should be compared with 7-10 existing method with additional parameters if possible.

Author Response

Response to Reviewer 2 Comments

Point 1: Introduction should focus on the adsorbent materials which should include different kind of adsorbents prepared form the same precursor materials, types of pollutants been studies, method of adsorbent synthesis etc.

Response 1: We added this paragraph in the introduction (see lines 52-56).

“According to previous research, several types of lignocellulosic agricultural have been tested for the adsorption removal of different pollutants [10-14]. To only cite few, date pits activated carbon [15], date palm biochar [16], nut shell-derived carbon [17], pistachio shells [18], edamame shell [19], peanut shell [20], etc”.

 Point 2: Authors were asked to check the references but seems it was not checked properly, reference 13 needs to be cross checked for citing reference for agricultural waste, this article deals in conducting polyaniline as adsorbent.

Response 2: We corrected all references, especially reference 13.

[13] Mahi, O.; Khaldi, K.; Belardja, M.S.; Belmokhtar, A.; Benyoucef. A. Development of a New Hybrid Adsorbent from Opuntia Ficus Indica NaOH Activated with PANI Reinforced and Its Potential Use in Orange G Dye Removal. Journal of Inorganic and Organometallic Polymers and Materials. 2021, 31, 2095-2104. https://doi.org/10.1007/s10904-020-01873-3

Point 3: Similarly refrence 19 in the article in Yusop, M.F.M. and co-worker not Ahmed et al.,

Response 3: We corrected the reference (see reference 33).

“Ahmad, M.A.; Ahmad, N.; Bello, O.S. Adsorptive removal of malachite green dye using durian seed-based activated carbon. J.Wat. Air. Soil Poll. 2014, 225, 2057-2064. https://doi.org/10.1007/s11270-014-2057-z.”

Point 4: Akar et al is not at reference number 30. The actual authors of the mentioned title are Mohd Azmier Ahmad, Norhidayah Ahmad & Olugbenga Solomon Bello.

Response 4: We corrected the reference, and its became under No 34 (see reference 34).

“Akar, E.; Altinişik, A.; Seki, Y. Using of activated carbon produced from spent tea leaves for the removal of malachite green from aqueous solution. Ecological Engineering. 2013, 52, 19-27. https://doi.org/10.1016/j.ecoleng.2012.12.032.”

Point 5: Table 5 needs to be enrich, at least it should be compared with 7-10 existing method with additional parameters if possible.

Response 5: We enriched the table 5 by references for comparison.

Reviewer 3 Report (Previous Reviewer 3)

Comments and Suggestions for Authors

row 18-19 : BET method is giving only the surface area and that is all. you should write nitrogen adsorption desorption isotherms or sorption isotherms

and please use the new iupac to mention the type of isotherms IVa (H4 seems your isotherm) from http://sol.rutgers.edu/~aneimark/PDFs/IUPAC_Report_PAC_2015.pdf

row 193 : should be written only TG not TGA, because TGA means that you have DTA and DTG aslo on graph where the processes can be seen, should be added at least DTA.

row 108 : the samples doesnt seems to be outgassed at 300 °C under vacuum for 24 h, they looked like they been degassed 1-2 hours maximum

row 125: check for typos there are a few like example equation 1 

Comments on the Quality of English Language

the english is better, could be improved.

Author Response

Response to Reviewer 3 Comments

Point 1: row 18-19 : BET method is giving only the surface area and that is all. you should write nitrogen adsorption desorption isotherms or sorption isotherms.

Response 1: We corrected the sentences, and they became as follows (see line 18-19)

“nitrogen adsorption desorption isotherms”.

 Point 2: and please use the new iupac to mention the type of isotherms IVa (H4 seems your isotherm) from http://sol.rutgers.edu/~aneimark/PDFs/IUPAC_Report_PAC_2015.pdf.

Response 2: Yes, according to the new IUPAC the type of isotherm is H4.

 Point 3: row 193 : should be written only TG not TGA, because TGA means that you have DTA and DTG aslo on graph where the processes can be seen, should be added at least DTA.

Response 3: We corrected the error to (TG) (see line 210)

Point 4: row 108 : The samples doesnt seems to be outgassed at 300 °C under vacuum for 24 h, they looked like they been degassed 1-2 hours maximum.

Response 4: Yes, there is a typo. The duration is 2 hours (see line 119)

Point 5: row 125: check for typos there are a few like example equation 1.

Response 5: We corrected the typo in Equation 1

This manuscript is a resubmission of an earlier submission. The following is a list of the peer review reports and author responses from that submission.

Round 1

Reviewer 1 Report

Comments and Suggestions for Authors

The manuscript, entitled “Optimization conditions of Malachite green adsorption onto carbon waste using D-Optimal experimental design” is not suit for publish in the journal Molecules.

This manuscript reports on the modified almond shell can be used as a sorbent for the adsorption of malachite green. First of all, the manuscript was not well designed and the writing style of the manuscript is very poor. Significant improvements in writing are required. The interpretations of the experimental observations were insufficient.

Title:

The title of the manuscript needs some improvements. Optimization conditions of Malachite green....... should be “Optimization conditions of malachite green......

Abstract:

1.Line17: Malachite green's adsorption onto almond shell as a cheap adsorbent was investigated. This sentence should be revised.

2.Line19: "The parameters starting dye concentration, adsorbent mass......" can be expressed in a more common way.

3.Line21: The Adsorption study was fitted to..... should be revised to “ The adsorption study was fitted to......

4.The abstract needs to be modified, and some sentences can be expressed more clearly. The title of the mauscript describes malachite green adsorbed by carbon materials, but in the abstract is the modified almond shell, is not clear.

 5.Keywords need to be reduced.

The introduction:

1.Line 31: Therefore, It is necessary to ...... should be “Therefore, it is necessary to .......

2.Line 33:  Filtration via membranes ..... should be filtration via membranes......

3.Lines 35-37: This sentence should be revised.

4.Lines 39-40: This sentence should be revised.

5.Lines 41-44: The references should be added.

6. Lines 45-46: AC is applied to polish the color, treat the odor and to remove heavy metals. This sentence is not clear, what is polish the color?

7.Lines 51:  Various applications [16]...the references are inadequate.

8. Lines 56-57: This sentence should be revised.

9.Line 60: Malachite Green (MG) is a cationic highly toxic dye.... this sentence should be revised.

10. Line 62: Malachite Green should be MG

11.Line 71:”MG was also purified using ceramic clays [34].” English must be clear.

12.The novelty statement should be added clearly in the Introduction.

13.The introduction section was not well designed. The use of the language is very poor. Generally, the introduction should be improved.

Materials and Methods:

 1. Line 89: Then, a graphite container was used to hold the carbon adsorbent.  whats this meaning? What is the purpose of holding carbon adsorbent in graphite containers?

2. Carbon and Charcoal are used alternately in this manuscript, please express it uniformly.

3.Lines 94-96:  X-ray diffraction was achieved in 2θ = 10–90° range, the step width was 0.03° and at step duration of 1s. By adsorbing N2, the texture of AC was characterized at 77 K (Quantachrome/Autosorb-6).  However, the XRD test result in the manuscript is not 10-90°.

3. Equation (1) needs to be modified. There are two equal signs.

4. Line 100: “M = 929.02 g.mole-1 should be “M = 929.02 g.mol-1 

5. Lines 120-122:  This part is repeated with Lines: 112-114!!!

6. Lines 115-119: This part is repeated with Lines: 123-127!!!

7. The part of  2.5. Adsorption Isotherms should be delete.

8. Lines 128-129:The linearised equation of Freundlich is wrong. It should be revised to 

9.Line 133: The unit of KL must be corrected.

10.The unit of KF must be corrected.

 Results and discussion:

1. Infrared and XRD in Figure. 2 have no ordinate, these two figures need to be redrawn.

2. TGA should be added to characterization, Why do you calcinate almond shell at 400 ° and not at other temperature, for example, 600 °?

3. SEM images of biosorbents should be added.

4. The X-axis notation in Figure 3(b) is incorrect. The Y-axis in Figure 3(b) has no ruler, please added it.

5.Lines 220:The equation (4) is wrong!!!

6. Line 217-219: It should be supported by references.

7. More experimental details about adsorption experiments should be added; For example: Effect of calcination temperature, the effect of initial concentration, and etc….

8. The fitting plots of Langmuir and Freundlich isotherm models should be added.

9.The effect of pH should be analysised before the absorbent dosage and adsorption isotherm. And the structure of the paper needs to be adjusted appropriately.

10.The fitting plots of pseudo-second-order and intraparticle diffusion kinetic models for MG adsorption should be added.

11.  Possible mechanism of MG adsorption should be added; this section should be added using some techniques including FTIR, XRD and XPS spectra before and after MG adsorption to determine the mechanism of dye removal.

Conclusion

1. The maximum adsorption capacity in the abstract is 166.66mg/g, while the maximum adsorption capacity in the conclusion is 146.45mg/g. In this manuscript, carbon derived from almond shell was used for adsorption, while in the conclusion is the almond shell. The same problem is found in the Introduction part.

 Before submitting a manuscript, the author should be sure that your manuscript has been properly prepared and formatted. Some equations in the manuscript are wrong!!! If the equations is wrong, the data of the paper is also wrong. The paper still needs extensive revision and is not available for publication in the MOLECULES.

Comments on the Quality of English Language

Moderate editing of English language required.

Reviewer 2 Report

Comments and Suggestions for Authors

1.       Abstract should be more informative

2.       Authors should improve the introduction and include details of water pollution, it should focus on the adsorbent material being studied in the manuscript

3.       Authors should check the reference e.g.  in the introduction section authors say reference 18 is the removal of pentachlorophenol, however, at 18th position in the reference list says @Almond shell bio composite hexavalent chromium ions and Orange G dye. This should be re-checked.

4.       Similarly, article at reference 17 says the paper is related to removal of malachite green and methyl red. Authors cited the same article as removal of Direct Red 80 dye.

5.       Authors should carefully read all the reference and cite them carefully.  Reference 19 in the list is removal of bromophenol blue not methylene blue reference 20 is Anionic Azo Dye not volatile organic compounds

6.       Page 2 line 85.,. In Mascara, Algeria, a local farm produces the almond shell; the sentence should be re written

7.       Section 2.4 How authors selected 0.1 g adsorbent for the study. Did they perform optimization study?

8.       Authors should mention in details about the pH studies.

9.       Page 3 Line 101. Authors should re check the concentration? Was 1g/L solution used throughout the experiment or some working solution was prepared. The same should be clearly mentioned in section 2.3.

10.   Page 5 line 168... “Absence” should be changed to “absence”

11.   Page 9 Line 263 as per eq 10 Yi should be Yi

12.   A comparison of the study with existing similar published results should be mentioned at appropriate place

  Reviewer 3 Report

Comments and Suggestions for Authors

The article used almond shell as a cheap adsorbent for Malachite green. The D-Optimal design was used as model for improvement of the adsorption.

The article has some errors that should be checked :

- In the introduction the authors should check more accurate the reference, for example check reference 22. But there are also some other information that need to be justified by relevant references. For example : - reference 34, row 71, should be complete, if is relevant for this study 

- please check the manuscript for typos , for example you have a lot of uppercase that need to be solved, but not only.

- the aim of this study should be rewritten in order to be more clear, row 79-82.

- please reformulate row 85- 86

- why did you filter, clean with acetone if you used thermal treatment at 400oC?

- row 165 :  82  ?  please add the ICDD card to see what type of carbon it was obtained.

- row175: Microstructure of charcoal based almond shell ?  why microstructure? - row  186 and row 19 ? different results !!!

- row 292 and row 23 ? different results !!!

- I suggest to the authors to make Raman studies in order to see what type of  carbon they obtained.